# Discrete Tensorized Label Learning with Anchor Graphs

## Abstract

Many discrete multi-view clustering methods based on anchor graphs use the anchor graph decomposition or spectral clustering to obtain the final clustering labels, such methods achieve good results but lack interpretability. Morever, some of them are poorly balanced. To this end, first, we start from the perspective of label transmission to convert labels of the anchors to the labels of the samples, which has better interpretability. Second, we find a new and remarkable use of the nuclear norm, i.e., maximizing the nuclear norm can ensure the balanced clusters, which has the rigorous theoretical proof. Simultaneously, a novel optimisation method based on the first order Taylor expansion is proposed for the nuclear norm. Finally, we introduce the tensor Schatten $p$-norm to fully exploit the spatial structural and complementary information between views, which can obtain aligned label matrices. Extensive experiments have verified the superiority of the proposed method compared with the state-of-the-art methods.

## 1 Introduction

Clustering as an unsupervised machine learning method aims at classifying unlabelled data into sets with specific meanings. This technique has been widely adopted across various fields due to its efficiency. In the early stages, most research focused on single-view data. However, the clustering results were often suboptimal due to incomplete information about the objects. In recent years, multi-view data, which allows obtaining object feature information from multiple perspectives, has garnered significant attention. Multi-view data provides a more complete and comprehensive representation of information. For example, a video can be described through text, images, voice, etc., and a news story can be translated into multiple languages such as Chinese, English, French, and more. As a result, multi-view clustering has developed rapidly and is now a prominent research area.

Numerous methods have been proposed for multi-view clustering, including subspace clustering, spectral clustering, and non-negative matrix factorization clustering. Among them, graph-based clustering methods have experienced significant development due to their excellent performance in capturing the spatial structure of samples, making them highly favored by researchers. The critical step in graph-based clustering is the construction of an affinity matrix (or similarity matrix) of size $N \times N$, where $N$ represents the number of samples. This matrix is used to represent the relationships between different sample points. Since similarities must be calculated for each sample, the time complexity of this process can reach $O(N^2)$, and the graph factorization process also has a high time complexity of $O(N^3)$. This results in substantial memory usage and processing time, especially for large datasets. To address this issue, some methods like Li et al. (2015) and Qiang et al. (2021) utilize anchor graph-based clustering. The anchor graph typically has a size of $N \times m$, where $m$ is the number of anchor points, and $m << N$, representing the relationship between anchor points and sample points. This approach significantly reduces the computational complexity, making it feasible to handle large-scale datasets efficiently.

The methods for processing multiple anchor graphs in multi-view clustering have become a hot research topic. Some approaches construct a unified graph for multi-view data instead of separate graphs for each view (Chen et al., 2020; Lin et al., 2021), while others construct individual graphs for each view and use multi-view learning to fuse them into a unified graph (Nie et al., 2017; Qiang et al., 2023; Zhan et al., 2018). Although these methods achieve promising clustering results, they rely on a unified "global graph" for each view, assuming the same spatial structure across views. In

reality, the hard labels and clustering results differ between views, and each view has a unique spatial structure. These methods fail to fully explore the spatial structures and complementary information across views.

Additionally, we observe that some graph-based or anchor graph-based methods (Kang et al., 2021; Liang et al., 2019; Wang et al., 2021) cannot directly obtain the final clustering results, often requiring post-processing methods such as k-means. These post-processing methods not only reduce efficiency but also constrain the clustering performance, limiting the ability to achieve optimal results. Moreover, we found that many multi-view clustering methods based on anchor graphs use the anchor graph decomposition or spectral clustering to obtain the final clustering labels, such as Yang et al. (2024) and Qiang et al. (2021), such methods achieve good results but lack interpretability. And, they are also unevenly balanced, making them less efficient for handling data with varied distributions.

To address these issues, this paper uses the label transmission strategy to design the clustering model by mining the relationship between the labels of anchors and the labels of samples, which has better interpretability. Besides, the introduce of the nuclear norm ensures that this model can obtain discrete and balanced clustering results directly, eliminating the need for time-consuming and performance-limiting post-processing methods. In addition, the tensor Schatten $p$-norm used in Gao et al. (2020); Yang et al. (2022); Lei et al. (2024); Lu et al. (2023); Li et al. (2023) explores low-rank representations between views, which can ensure that each graph retains the spatial structural information of its corresponding view without distortion. Our contributions are summarized as follows:

- We propose a label transmission strategy to establish the relationship between anchor points and sample points based on an anchor graph, so that the anchor labels can be transferred to the sample labels, which has more interpretable than previous methods.

- Different from the previous approach of minimizing the nuclear norm, we propose the maximization of the nuclear norm to ensure the discretised and class-equilibrated in the clustering process, which has the rigorous theoretical proof.

- For the optimization of the nuclear norm, we propose an innovative and highly efficient optimization approach grounded in the first-order Taylor expansion.

- We introduce the tensor Schatten $p$-norm to fully explore the spatial complementary information between views. This can ensure that labels of samples from different views are more likely to align, thereby enhancing clustering performance.

## 2 RELATED WORK

### 2.1 GRAPH-BASED CLUSTERING

There are a number of methods that perform weighted embedding of multiple graphs to obtain fused graphs. Liang et al. (2019; 2022), by exploiting the consistency and inconsistency between individual views for multi-view graph learning to obtain a unified graph common to all views, the authors of these two methods believe that there are consistent as well as inconsistent parts between the views, and thus design an objective function so that the model can learn the consistency between views to obtain a common graph of fused terms. Wang et al. (2019) recognises that the importance of the views is not consistent, and thus chooses to fuse multiple graphs by weighting them thus obtaining a consistent representation. Qiang et al. (2023) proposed a very concise graph-based clustering method using an objective function with only one item in order to obtain a weighted multi-view aggregated graph with good results.

Even though these methods give good clustering results, multi-graph fusion is not always a good choice for multi-view. Fusion of graphs involves treating the hard labels in each view as consistent, but this is not reasonable. The clustering results of different views will often not be identical, and methods that directly use the results of multi-graph fusion to obtain hard labels do not fully exploit the complementary information between views and ignore the spatial structure information of each view. At the same time, we also note that these methods use full-sample graphs, and the computational complexity of this construction process and decomposition process is $O(N^2)$ or even $O(N^3)$, especially on large-scale data this problem is particularly significant.

## 2.2 ANCHOR GRAPH-BASED CLUSTERING

In order to reduce the computational complexity, methods such as Li et al. (2015) propose an anchor graph-based clustering approach. The number of anchor points $m$ used to construct the anchor graph is much smaller than the number of samples $N$ ($m << N$), which can significantly reduce the space and time complexity of the model. Similarly, in the anchor graph-based approach, Kang et al. (2021) proposed a framework that can be applied to both single and multiple views. They similarly recognise that different views may play different roles and choose to weight different views, but the key graph remains applicable across views using a uniform graph. Yang et al. (2024); Qiang et al. (2021); Shi et al. (2021), on the other hand, weights the anchor graphs for different views according to their importance and fuses the anchor graphs for each view to get a unified graph. These three discrete clustering methods do not fully explore the spatial structure information and complementary information between views. In addition, they all use matrix decomposition to obtain the final labelling matrix, which makes the interpretability worse. Also in addition to this, these methods above they try to make the hard labels consistent for each view clustering, this requirement is too high and different from the actual situation, resulting in clustering results are not good.

## 3 METHODOLOGY

### 3.1 MOTIVATION AND OBJECTIVE

Before proposing our model, we introduce Theorem 3.1 first.

**Theorem 3.1.** *Given $n_1 + n_2 + \cdots + n_C = N$, where $n_k \geq 0$ denotes the number of samples in the $k$-th cluster, the optimal value of problem (1) can be obtained when $n_1 = n_2 = \cdots = n_C = \frac{N}{C}$, which equals $\sqrt{NC}$. At this point, the probability matrix $\mathbf{U}$ is discrete and shows a balanced class distribution.*

$$\max_{\mathbf{U} \geq 0, \mathbf{U}\mathbf{1}=\mathbf{1}} \parallel \mathbf{U} \parallel_*, \tag{1}$$

*where, $\parallel \cdot \parallel_*$ is the matrix nuclear norm.*

*Proof.* Let $\sigma_i(\cdot)$ denote the $i$-th largest singular value of a matrix, and $\tau_i(\cdot)$ denote the $i$-th largest eigenvalue of a matrix, we have

$$\parallel \mathbf{U} \parallel_* = \sum_{i=1}^{C} \sigma_i(\mathbf{U}) = \sum_{i=1}^{C} \sqrt{\tau_i(\mathbf{U}^T\mathbf{U})} = tr\sqrt{\mathbf{U}^T\mathbf{U}}, \tag{2}$$

where, $\sigma_i^2(\mathbf{U}) = \tau_i(\mathbf{U}^T\mathbf{U})$, and $tr(\mathbf{U}^T\mathbf{U}) = \sum_{i=1}^{C} \tau_i(\mathbf{U}^T\mathbf{U})$.

Thus, we can obtain the following equivalent form of problem (2):

$$\max_{\mathbf{U} \geq 0, \mathbf{U}\mathbf{1}=\mathbf{1}} \parallel \mathbf{U} \parallel_* \Leftrightarrow \max_{\mathbf{U} \geq 0, \mathbf{U}\mathbf{1}=\mathbf{1}} tr(\mathbf{U}^T\mathbf{U}) = \max_{\mathbf{U} \geq 0, \mathbf{U}\mathbf{1}=\mathbf{1}} \sum_{i=1}^{N} \sum_{j=1}^{C} u_{ij}^2. \tag{3}$$

Since the samples in $\mathbf{U}$ are independent, each row is independent. The problem of the $i$-th row is $\max_{\mathbf{u}^i\mathbf{1}=1} \sum_{j=1}^{C} u_{ij}^2$, whose optimal solution is that $\mathbf{u}^i$ has one and only one element equal to 1, with the rest of the elements being 0. Therefore, the maximum value of the problem (3) is $N$, leading to the conclusion that maximizing the nuclear norm of the probability matrix forces the discrete label matrix.

In this case, the matrix $\mathbf{U}^T\mathbf{U} \in \mathbb{R}^{C \times C}$ is a diagonal matrix, where the $j$-th element $n_j$ equals the number of samples contained in the $j$-th cluster. Therefore, we have

$$\parallel \mathbf{U} \parallel_* = \sum_{j=1}^{C} \sqrt{n_j}. \tag{4}$$

Further, we define $\boldsymbol{u} = [\sqrt{n_1}, \ldots, \sqrt{n_C}]^T \in \mathbb{R}^{C \times 1}$ and $\mathbf{q} = [1, \ldots, 1]^T \in \mathbb{R}^{C \times 1}$. According to the Cauchy-Schwarz inequality, $|\langle \boldsymbol{u}, \boldsymbol{q} \rangle| \leq \|\boldsymbol{u}\|_2 \|\boldsymbol{q}\|_2$, we have

$$\sqrt{n_1} + \cdots + \sqrt{n_C} \leq \sqrt{n_1 + \cdots + n_C}\sqrt{1 + \cdots + 1} = \sqrt{NC}. \tag{5}$$

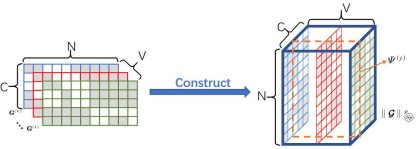

Figure 1: Construction of tensor $\mathcal{G} \in \mathbb{R}^{N \times V \times C}$, where $\boldsymbol{\Psi}^{(j)}(j \in \{1, 2, \ldots, N\})$ denotes the $j$-th frontal slice of tensor $\mathcal{G}$.

Equality in (5) holds if and only if $n_1 = \cdots = n_C = N/C$. $\qquad\square$

Inspired by the method of constructing anchor graphs to reduce computational complexity, we propose a label transmission strategy. Specifically, $\mathbf{S} \in \mathbb{R}^{N \times m}$ represents the correlation between the sample points and the anchor points—the larger the value, the higher the correlation, with the relationship being reciprocal. We can consider $\mathbf{S}$ as the similarity between anchor points and sample points, and $\mathbf{G} \in \mathbb{R}^{N \times C}$ as the similarity between sample points and categories. We consider $\mathbf{Z} = \{z_i\}_{i=1}^N$ and $\mathbf{A} = \{a_j\}_{j=1}^m$ are the sets consisting of sample points and anchor points, respectively. The probabilistic transmission process is considered as a weighting of similarity. The one-step transition probability matrix $T$ can be computed as $T = (O^S)^{-1}S$, where $o_{ii}^S = \sum_{j=1}^{N+m} s_{ij}$. Thus, the transition probability from the $i$-th sample to the $j$-th anchor point is given by Liu et al. (2010)

$$t^{(1)}(a_j \mid z_i) = \frac{s_{ij}}{\sum_{j'=1}^m s_{ij'}} = s_{ij}. \tag{6}$$

Similarly, let $g_{ik}$ denote the weight between sample point $x_i$ and class $c_k$. The transition probability from the $i$-th sample point to the $k$-th class is

$$t^{(1)}(c_k \mid z_i) = \frac{g_{ik}}{\sum_{k'=1}^c g_{ik'}} = g_{ik}. \tag{7}$$

The probability transfer process can be treated as a weighted sum of similarities. Finally, the transition probability from anchor points to categories is

$$t(c_k \mid a_j) = t^{(1)}(z_i \mid a_j)t^{(1)}(c_k \mid z_i) = \sum_{i=1}^N s_{ij}g_{ik}. \tag{8}$$

By using a label transmission strategy with anchors, we can better handle large-scale data. Considering the multi-view clustering problem, we denote $\mathbf{G}_v \in \mathbb{R}^{N \times C}$ as the label matrix of the $v$-th view. We then let $\mathbf{G}$ be an indicator matrix. Thus, we propose the following model:

$$\max_{\mathbf{G}_v \in Ind} \sum_{v=1}^V \| \mathbf{S}_v^T \mathbf{G}_v \|_*, \tag{9}$$

where, $S_v \in \mathbb{R}^{N \times m}$ is the anchor graph of the $v$-th view. $Ind$ refers to an indicator matrix where only one element in each row is 1, and the remaining elements are 0. $\mathbf{G}_v$ denotes the $v$-th view of tensor $\mathcal{G}$, which is constructed as shown in Fig. 1.

Constraining the solution within a single view alone does not sufficiently guarantee that the corresponding samples in each view will be grouped into the same clusters. Since we are addressing a multi-view problem, the matrices from each view form a tensor. Thus, we introduce the tensor Schatten $p$-norm as a regularization term.

**Definition 3.2.** (Gao et al., 2020) Given a tensor $\mathcal{M} \in \mathbb{R}^{n_1 \times n_2 \times n_3}$, where $h = \min(n_1, n_2)$, the tensor Schatten $p$-norm of $\mathcal{M}$ is defined as:

$$\|\mathcal{M}\|_{\textcircled{S}p} = \left( \sum_{i=1}^{n_3} \|\overline{\mathcal{M}}^{(i)}\|_{\textcircled{S}p}^p \right)^{\frac{1}{p}} = \left( \sum_{i=1}^{n_3} \sum_{j=1}^h \sigma_j(\overline{\mathcal{M}}^{(i)})^p \right)^{\frac{1}{p}}, \tag{10}$$

where, $\sigma_j(\overline{\mathcal{M}}^{(i)})$ represents the $j$-th singular value of $\overline{\mathcal{M}}^{(i)}$.

*Remark* 3.3. When $p = 1$, the tensor Schatten $p$-norm of $\mathcal{M} \in \mathbb{R}^{n_1 \times n_2 \times n_3}$ becomes the tensor nuclear norm (Gao et al., 2020), which is defined as:

$$\|\mathcal{M}\|_* = \sum_{i=1}^{n_3} \sum_{j=1}^{h} \sigma_j \left( \overline{\mathcal{M}}^{(i)} \right).$$

Let us take the matrix Schatten $p$-norm as an example. We have $\|\mathbf{M}\|_{\mathcircled{S}p}^p = \sigma_1^p + \cdots + \sigma_h^p$, where $p > 0$, $\mathbf{M} \in \mathbb{R}^{n_1 \times n_2}$, and $\sigma_i$ denotes the $i$-th singular value of $\mathbf{M}$. As shown in Nie et al. (2012), $\lim_{p \to 0} \|\mathbf{M}\|_{\mathircled{S}p}^p = \#\{i : \sigma_i \neq 0\} = \text{rank}(\mathbf{M})$.

As illustrated in Fig. 1, for tensor $\mathcal{G}$, the $j$-th frontal slice $\boldsymbol{\Psi}^{(j)}$ represents the relationship between the $N$ samples and the $j$-th cluster in different views. Moreover, in practice, the structure of clusters varies significantly across views. The Schatten $p$-norm constraint on $\mathcal{G}$ ensures that the $j$-th slice $\boldsymbol{\Psi}^{(j)}$ exhibits a spatially low-rank structure. By minimizing the Schatten $p$-norm, the clustering results within each view can be made more consistent, enabling better utilization of the complementary information between views.

Thus, the problem (9) can be improved to the following form:

$$\max_{\mathbf{G}_v \in Ind} \sum_{v=1}^{V} \|\mathbf{S}_v^T \mathbf{G}_v\|_* - \lambda \|\mathcal{G}\|_{\mathircled{S}p}^p, \tag{11}$$

where, $\lambda$ and $p$ are hyperparameters.

In general, different views in multi-view data often contain various heterogeneous features, and each view does not contribute to a specific task to the same extent. The fusion of multi-view information has a significant impact on the effectiveness of multi-view learning, as each view contributes differently for clustering purposes. We choose to apply learnable weights, allowing the model to learn the weights between views during the optimization process, assigning them to view-specific anchor clustering results. The final objective function is formulated as follows:

$$\max_{\mathbf{G}_v, \alpha_v} \sum_{v=1}^{V} \alpha_v^r \|\mathbf{S}_v^T \mathbf{G}_v\|_* - \lambda \|\mathcal{G}\|_{\mathircled{S}p}^p, \quad \text{s.t. } \mathbf{G}_v \in Ind, \alpha_v \geq 0, \sum_{v=1}^{V} \alpha_v = 1, \tag{12}$$

where, $0 \leq \alpha_v \leq 1$ is the weight of the $v$-th view, and $r$ is a hyperparameter.

### 3.2 OPTIMIZATION

Since the nuclear norm in (12) involves the sum of the singular values of a matrix, which is not always a smooth function, direct optimization of problems involving nuclear norm tends to be complex and difficult.

In this case, we can take $f(\mathbf{X}) = \|\mathbf{S}_v^T \mathbf{G}_v\|_* = \|\mathbf{X}\|_*$, and perform a first-order Taylor expansion at the solution $\mathbf{X}^{(t)}$ obtained in the $t$-th iteration, i.e.,

$$f(\mathbf{X}) = f(\mathbf{X}^{(t)}) + \langle \nabla f(\mathbf{X}^{(t)}), \mathbf{X} - \mathbf{X}^{(t)} \rangle, \tag{13}$$

where, $\nabla f(\mathbf{X}^{(t)})$ is the gradient of $f(\mathbf{X}) = \| \cdot \|_*$ at $\mathbf{X}^{(t)}$. Based on SVD and according to Zhen et al. (2017), we have

$$\mathbf{H} = \nabla f(\mathbf{X}) = \frac{\partial \|\mathbf{X}\|_*}{\partial \mathbf{X}} = \mathbf{U} \boldsymbol{\Sigma}^{-1} |\boldsymbol{\Sigma}| \mathbf{V}^T, \tag{14}$$

where, $\mathbf{X} = \mathbf{U} \boldsymbol{\Sigma} \mathbf{V}^T$ and $\boldsymbol{\Sigma}^{-1}$ is the Moore-Penrose pseudo-inverse of $\boldsymbol{\Sigma}$. At this point, if we ignore the constants in Eq. (13), we obtain the following form:

$$\langle \nabla f(\mathbf{X}^{(t)}), \mathbf{X} \rangle = \text{tr}(\nabla f(\mathbf{X}^{(t)})^T \mathbf{X}) = \text{tr}(\mathbf{H}^T \mathbf{X}). \tag{15}$$

Up to this point, the nuclear norm in the antecedent term of model (12) gets an equivalent form. For the Schatten $p$-norm in the latter term, we use the Augmented Lagrange Multiplier (ALM) method

and introduce the auxiliary variable $\mathcal{J}$, such that $\mathcal{J} = \mathcal{G}$. Model (12) then leads to the following form:

$$\max_{\alpha_v, \mathbf{G}_v, \mathcal{J}} \sum_{v=1}^{V} \alpha_v^r \mathrm{tr}(\mathbf{H}_v^T \mathbf{S}_v^T \mathbf{G}_v) - \lambda \|\mathcal{J}\|_{\circledS}^p - \frac{\mu}{2} \|\mathcal{G} - \mathcal{J} + \frac{\mathcal{M}}{\mu}\|_F^2,$$ (16)

$$\text{s.t. } \mathbf{G}_v \in Ind, \alpha_v \geq 0, \sum_{v=1}^{V} \alpha_v = 1,$$

where, $\mathcal{M}$ is the Lagrange multiplier and $\mu$ is the penalty parameter.

To solve this problem, we divide it into four steps.

- **Solve $\mathcal{G}$ with fixed $\mathcal{J}$.** The model (12) becomes:

$$\max_{\mathbf{G}_v \in Ind} \sum_{v=1}^{V} \alpha_v^r \, \mathrm{tr}(\mathbf{H}_v^T \mathbf{S}_v^T \mathbf{G}_v) - \frac{\mu}{2} \sum_{v=1}^{V} \|\mathbf{G}_v - \mathbf{J}_v + \frac{\mathbf{M}_v}{\mu}\|_F^2.$$ (17)

Expanding the posterior term and ignoring constants, we obtain the following form:

$$\max_{\mathbf{G}_v \in Ind} \sum_{v=1}^{V} \mathrm{tr} \left( \mathbf{G}_v^T \left( \alpha_v^r \mathbf{S}_v \mathbf{H}_v + \mu \mathbf{J}_v - \mathbf{M}_v \right) \right).$$ (18)

Since the views are independent, within each view, $\mathbf{G}_v$ is updated by:

$$\mathbf{G}_{ib} = \begin{cases} 1, & \arg\max_{i}(\alpha_v^r \mathbf{S}_v \mathbf{H}_v + \mu \mathbf{J}_v - \mathbf{M}_v)_i, \\ 0, & \text{otherwise,} \end{cases}$$ (19)

where, $i = 1, 2, \ldots, N$ and $b = 1, 2, \ldots, C$.

- **Solve $\mathcal{J}$ with fixed $\mathcal{G}$.** The model (12) becomes:

$$\min_{\mathcal{J}} \lambda \|\mathcal{J}\|_{\circledS}^p + \frac{\mu}{2} \|\mathcal{G} - \mathcal{J} + \frac{\mathcal{M}}{\mu}\|_F^2.$$ (20)

This problem has a closed-form solution as presented in Theorem 3.4 from Gao et al. (2020).

**Theorem 3.4.** *Let $\mathcal{A} \in \mathbb{R}^{n_1 \times n_2 \times n_3}$ have a t-SVD $\mathcal{A} = \mathcal{U} * \mathcal{S} * \mathcal{V}^T$. For the model:*

$$\arg\min_{\mathcal{X}} \frac{1}{2} \|\mathcal{X} - \mathcal{A}\|_F^2 + \tau \|\mathcal{X}\|_{\circledS}^p,$$ (21)

*the global optimal solution is given by:*

$$\mathcal{X}^* = \mathbf{\Gamma}_\tau(\mathcal{A}) = \mathcal{U} * \mathrm{ifft}(\mathbf{P}_\tau(\bar{\mathcal{A}})) * \mathcal{V}^T,$$ (22)

*where, $\mathbf{P}_\tau(\bar{\mathcal{A}}) \in \mathbb{R}^{n_1 \times n_2 \times n_3}$ is obtained by the General Shrinkage Thresholding algorithm, $\bar{\mathcal{A}}$ is the Fourier transform of $\mathcal{A}$, and $\mathbf{\Gamma}_\tau(\mathcal{A})$ is the t-SVD of $\mathcal{A}$.*

Thus, the solution to problem (20) is:

$$\mathcal{J}^* = \mathbf{\Gamma}_{\frac{\lambda}{\mu}} \left( \mathcal{G} + \frac{\mathcal{M}}{\mu} \right).$$ (23)

- **Solve $\alpha$ with fixed $\mathcal{G}$ and $\mathcal{J}$.** The model (12) becomes:

$$\max_{\alpha_v} \sum_{v=1}^{V} \alpha_v^r \, \mathrm{tr}(\mathbf{H}_v^T \mathbf{S}_v^T \mathbf{G}_v), \quad \text{s.t.} \quad \alpha_v \geq 0, \sum_{v=1}^{V} \alpha_v = 1.$$ (24)

Using the method of Lagrange multipliers, we define the following Lagrange function:

$$\mathcal{L}(\alpha_1, \alpha_2, \ldots, \alpha_v, \gamma) = \sum_{v=1}^{V} \alpha_v^r \mathbf{N}_v + \gamma \left( 1 - \sum_{v=1}^{V} \alpha_v \right) + \sum_{v=1}^{V} \mu_v(-\alpha_v),$$ (25)

where, $\mu_v$ and $\gamma$ are the Lagrange multipliers, and $\mathbf{N}_v = \mathrm{tr}(\mathbf{H}_v^T \mathbf{S}_v^T \mathbf{G}_v)$. The optimal solution to problem (25) is given by Lu et al. (2023):

$$\alpha_v = \frac{\mathbf{N}_v^{\frac{1}{1-r}}}{\sum_{v=1}^V \mathbf{N}_v^{\frac{1}{1-r}}}, \tag{26}$$

where, $\mathbf{N}_v = \mathrm{tr}(\mathbf{H}_v^T \mathbf{S}_v^T \mathbf{G}_v)$.

- **Update $\mu$ and $\mathcal{M}$.**

  The penalty parameter $\mu$ is updated by

$$\mu = \min(\rho\mu, \mu_{max}), \tag{27}$$

where $\rho = 1.1$ and $\mu_{max} = 10^9$. The Lagrange multiplier $\mathcal{M}$ is updated by

$$\mathcal{M} = \mathcal{M} + \mu(\mathcal{G} - \mathcal{J}). \tag{28}$$

The pseudo-code for the entire optimization algorithm is summarized as Algorithm 3.1.

---

**Algorithm 3.1** Optimization to problem (12)

---

1: **Input** Data matrixes $\{\mathbf{D}^{(v)}\}_{v=1}^V \in \mathbb{R}^{N \times d_v}$, anchor rate $\theta$.
2: **Initialize** $p, \lambda, r, \mathcal{G}, \mathcal{J} = \mathcal{M} = \mathbf{0}$, S, $\alpha_v = \frac{1}{V}, (v = 1, \ldots, V), \rho = 1.1, \mu = 10^{-5}, \mu_{max} = 10^9$.
3: Contrust the anchor graph S by Xia et al. (2022).
4: **while** not convergence **do**
5:     Update $\mathcal{G}$ by (19);
6:     Update $\mathcal{J}$ by (23);
7:     Update $\mathcal{M}$ by $\mathcal{M} = \mathcal{M} + \mu(\mathcal{G} - \mathcal{J})$;
8:     Update $\alpha_v$ by (26), $(v = 1, \ldots, V)$;
9:     Update $\mu$ by $\mu = \min(\rho\mu, \mu_{max})$;
10: **end while**
11: **Output** indicator matrix $\{\mathbf{G}_v\}_{v=1}^V \in Ind$.

---

## 4 EXPERIMENTS

In this section, we compare the proposed anchor graph-based multi-view clustering algorithm with nine state-of-the-art multi-view clustering methods.

### 4.1 DATASETS

We conducted experiments on some real-world datasets: MSRC (Winn & Jojic, 2005), HandWritten (Asuncion et al., 2007), Mnist (Deng, 2012), Scene (Oliva & Torralba, 2001), NoisyMnist (Wang et al., 2015), and NUS-WIDE (Chua et al., 2009). The relevant information for each dataset is shown in Table 2 in the Appendix.

### 4.2 IMPLEMENTATIONS

All experiments were performed on an Intel Xeon Platinum 8168 CPU running the Windows 10 operating system, using MATLAB R2020b. For the anchor rate when constructing the anchor graphs, we adjusted it from 0.1 to 1 in steps of 0.1 for all datasets except the two largest ones. For the parameter $p$ in the tensor Schatten $p$-norm, we similarly varied it from 0.1 to 1 in steps of 0.1. The weights $\lambda$ of the tensor Schatten $p$-norm were divided into two parts, adjusted from 0.1 to 1 in steps of 0.1, and from 2 to 20 in steps of 1. The parameter $r$ for the weights of the views $\alpha_v$ was also divided into two parts: adjusted from 0.1 to 0.9 in steps of 0.1, and from 2 to 20 in steps of 1.

### 4.3 RESULTS AND ANALYSIS

*Comparative algorithms*

Table 1: Clustering performance of different method on real world datasets.

| Dataset | MSRC | | | HW | | | Mnist | | | Scene | | | NoisyMnist | | | NUS-WIDE | | |
|---|---|---|---|---|---|---|---|---|---|---|---|---|---|---|---|---|---|---|
| Metric | ACC | NMI | Purity | ACC | NMI | Purity | ACC | NMI | Purity | ACC | NMI | Purity | ACC | NMI | Purity | ACC | NMI | Purity |
| SMSC | 0.766 | 0.717 | 0.804 | 0.742 | 0.781 | 0.759 | 0.913 | 0.789 | 0.913 | 0.508 | 0.535 | 0.576 | OM | OM | OM | OM | OM | OM |
| SwMC | 0.776 | 0.774 | 0.805 | 0.758 | 0.833 | 0.792 | 0.914 | 0.799 | 0.912 | - | - | - | OM | OM | OM | - | - | - |
| LMVSC | 0.814 | 0.717 | 0.814 | 0.904 | 0.831 | 0.940 | 0.892 | 0.726 | 0.892 | 0.561 | 0.512 | 0.581 | 0.388 | 0.344 | 0.434 | 0.133 | 0.110 | 0.167 |
| FPMVS-CAG | 0.843 | 0.738 | 0.843 | 0.850 | 0.787 | 0.850 | 0.887 | 0.719 | 0.887 | 0.541 | 0.584 | 0.541 | 0.554 | 0.513 | 0.567 | 0.174 | 0.128 | 0.210 |
| CSMSC | 0.682 | 0.767 | 0.862 | 0.806 | 0.793 | 0.867 | 0.643 | 0.601 | 0.728 | 0.576 | 0.574 | 0.629 | OM | OM | OM | OM | OM | OM |
| GMC | 0.895 | 0.809 | 0.895 | 0.879 | 0.882 | 0.879 | 0.921 | 0.807 | 0.921 | 0.409 | 0.430 | 0.417 | - | - | - | - | - | - |
| ETLMSC | 0.962 | 0.937 | 0.962 | 0.938 | 0.894 | 0.938 | 0.934 | 0.847 | 0.934 | 0.218 | 0.166 | 0.221 | OM | OM | OM | OM | OM | OM |
| MSC-BG | 0.981 | 0.960 | 0.981 | 0.889 | 0.922 | 0.889 | 0.938 | 0.861 | 0.938 | 0.519 | 0.602 | 0.562 | - | - | - | - | - | - |
| Orth-NTF | 0.990 | 0.978 | 0.990 | 0.985 | 0.969 | 0.985 | 0.977 | 0.926 | 0.977 | 0.758 | 0.804 | 0.759 | 0.605 | 0.593 | 0.627 | 0.355 | 0.544 | 0.465 |
| Ours | **1.000** | **1.000** | **1.000** | **1.000** | **1.000** | **1.000** | **0.999** | **0.999** | **0.999** | **0.988** | **0.977** | **0.988** | **0.740** | **0.792** | **0.793** | **0.528** | **0.554** | **0.528** |

- SMSC (Hu et al., 2020): Combines non-negative embedding and spectral embedding. Non-negative embedding is learned to obtain the final clustering results directly, avoiding performance degradation due to post-processing.

- SwMC (Nie et al., 2017): A unified graph is obtained by weighted fusion of different views, minimizing the difference between a common indication matrix and the similarity matrix.

- LMVSE (Kang et al., 2020): Fuses the graphs of all views to get a unified graph and uses spectral clustering for the final result.

- FPMVS-CAG (Wang et al., 2021): Unifies anchor learning with anchor graph construction, optimizing jointly with linear time complexity, followed by post-processing to obtain the final clustering results.

- CSMSC (Luo et al., 2018): The self-representation matrix is divided into consistent and exclusive parts, with the former reflecting low-rank results and the latter highlighting the unique variations of each view.

- GMC (Wang et al., 2019): Fuses multiple graphs by weighting them to obtain a consistent representation.

- ETLMSC (Wu et al., 2019): Constructs a probability matrix from the tensor, with spectral clustering providing the final result.

- MSC-BG (Yang et al., 2022): Constrains two-part graphs using tensor Schatten $p$-norms to efficiently capture spatial structure and complementary information between views.

- Orth-NTF (Li et al., 2024): Applies orthogonal non-negative tensor factorization with tensor Schatten $p$-norm constraints.

The experimental results of the nine comparison algorithms on six datasets are shown in Table 1. Our method consistently achieves promising results across all datasets, validating its effectiveness.

- In summary, our proposed model outperforms the comparison algorithms in most cases.

- Additionally, we obtain clustering results directly without post-processing, outperforming algorithms such as SMSC, LMVSE, FPMVS-CAG, and ETLMSC that rely on post-processing for final results. This demonstrates better stability compared to methods requiring post-processing.

- Compared to methods like SwMC, LMVSE, and GMC, which fuse views into a unified common graph, our method preserves the anchor graph for each view, fully leveraging spatial structure and complementary information using tensor Schatten $p$-norms, resulting in improved performance.

- Although MSC-BG also employs tensor Schatten $p$-norm constraints, our approach accounts for the uniqueness of different views by preserving each view's individual graph.

- While Orth-NTF uses non-negative matrix factorization with tensor Schatten $p$-norm constraints, our approach leverages anchor graphs to explore the spatial structure relationship between anchor and sample points more effectively.

## 4.4 PARAMETER SENSITIVITY ANALYSIS

**Effect of anchor rate:** To evaluate the influence of the anchor rate on clustering performance, we used multiple clustering metrics and conducted experiments on several datasets. The results are

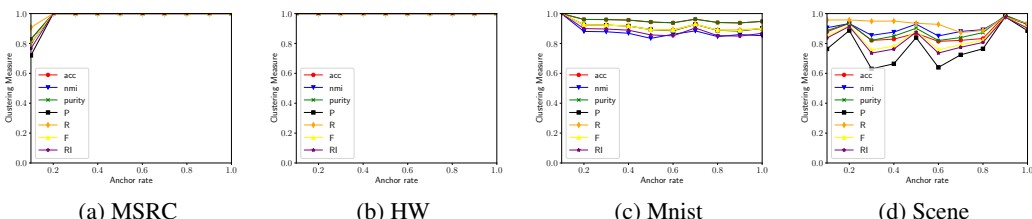

(a) MSRC       (b) HW       (c) Mnist       (d) Scene

Figure 2: The performance of the proposed method with different anchor rates

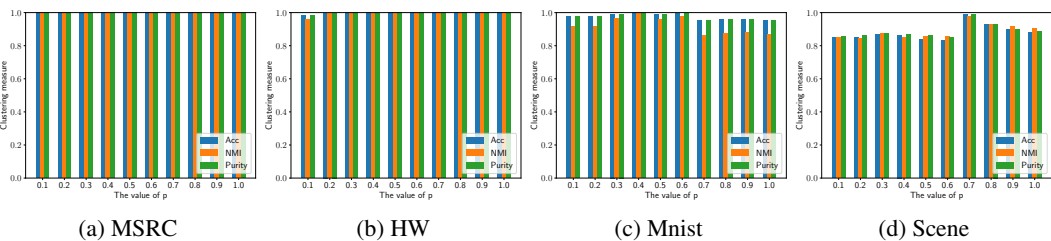

(a) MSRC       (b) HW       (c) Mnist       (d) Scene

Figure 3: The performance of the proposed method with different $p$

visualized in Fig. 2. As described earlier, we varied the anchor rate from 0.1 to 1 in increments of 0.1. Except for the MSRC dataset, the clustering performance did not show a consistent improvement as the anchor rate increased. Furthermore, the results remained relatively stable despite changes in the anchor rate.

**Effect of tensor Schatten $p$-norm:** We investigated the effect of the parameters of the tensor Schatten $p$-norm on the clustering results across multiple datasets. As mentioned above, we varied p from 0.1 to 1 in increments of 0.1 and observed Acc, NMI, and Purity for each dataset, which we visualized as a histogram in Fig. 3. It can be seen that the parameter p does not have a significant impact on individual datasets.

**Effect of $\lambda$:** The histogram depicting the clustering performance with varying $\lambda$ is presented in Fig. 4. We adjusted $\lambda$ over two segments and displayed the results for one range of [0.1, 0.5, 1, 5, 10, 15, 20]. Except for the Mnist dataset, it can be observed that the clustering performance improves progressively with increasing $\lambda$. This parameter aims to balance the contributions of the two terms in the objective function, which generally do not equalize at the same value across different datasets.

**Effect of r:** Fig. 5 shows a histogram of the clustering results as a function of the hyperparameter r. We tested one series $r = [-1, 0.001, 0.01, 0.1, 0.5, 0.9, 2, 5, 10, 20]$. Except for the Mnist dataset, the best clustering results were achieved with smaller values of r. This is because, at larger values of r, an exponential explosion occurs, making the weights nearly zero and leading to poorer results.

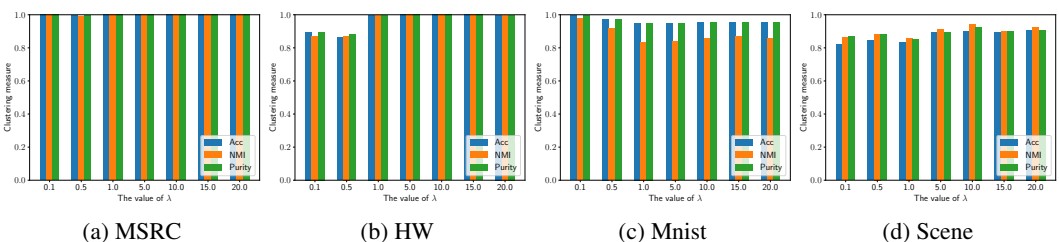

(a) MSRC       (b) HW       (c) Mnist       (d) Scene

Figure 4: The performance of the proposed method with different $\lambda$

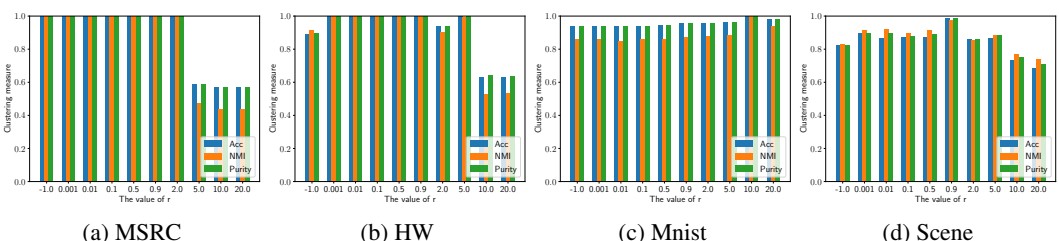

|          (a) MSRC          |          (b) HW           |          (c) Mnist          |          (d) Scene          |

Figure 5: The performance of the proposed method with different r

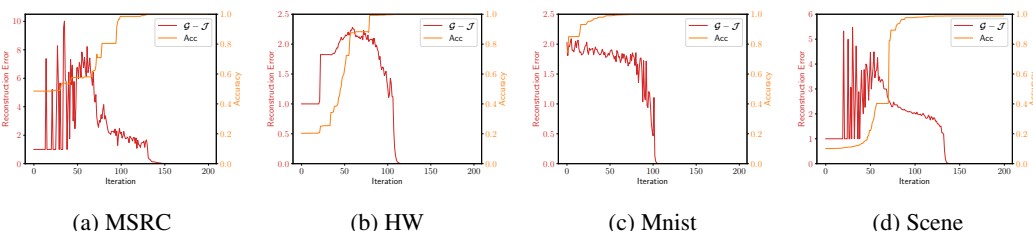

|          (a) MSRC          |          (b) HW           |          (c) Mnist          |          (d) Scene          |

Figure 6: The convergence of the proposed method

### 4.5 CONVERGENCE STUDY

Fig. 6 presents the curve of the clustering effect and the convergence discriminant condition with increasing iterations. The discriminant used is the reconstruction error (i.e., $\|\mathcal{G} - \mathcal{J}\|_\infty$), while the clustering effect is measured by Acc. It can be observed that our algorithm generally converges within 150 iterations.

## 5 CONCLUSION

In this paper, we propose a multi-view clustering method based on anchor graphs, which directly obtains the final clustering results. Specifically, the label transmission strategy is proposed to capture the relationship between samples and anchors, which has better interpretability. Moreover, maximizing the nuclear norm can ensure the balanced clusters, followed by a novel optimization method. In addition, by utilizing tensor Schatten $p$-norm, it fully exploits the spatial structural information and the complementary information between views to enhance clustering performance. The contribution of each view is reassessed, and individual views are weighted according to their significance. Experiments conducted on four standard datasets and two large-scale datasets demonstrate the excellent performance of the proposed algorithm, affirming its superiority. Additionally, a series of parametric and convergence analysis experiments are presented to illustrate the robustness and effectiveness of the method.

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

# A APPENDIX

## A.1 MORE DETIALS OF EXPERIMENTS

Table 2: Statistics of real world datasets

| Dataset | Instances | Views | Classes | Dimension |
|---------|-----------|-------|---------|-----------|
| MSRC | 210 | 5 | 7 | 24/576/512/256/254 |
| HW | 2000 | 4 | 10 | 76/216/47/6 |
| Mnist | 4000 | 3 | 4 | 30/9/30 |
| Scene | 4485 | 3 | 15 | 1800/1180/1240 |
| NoisyMnist | 50000 | 2 | 10 | 784/784 |
| NUS-WIDE | 30000 | 5 | 31 | 64/225/144/73/128 |

