# OpenReview forum: "Discrete Tensorized Label Learning with Anchor Graphs"
_ICLR.cc/2025/Conference — ICLR 2025 Conference Withdrawn Submission_

### Official Review · Reviewer_JjiR · 2024-10-21

**Soundness:** 2
**Presentation:** 2
**Contribution:** 2
**Rating:** 5
**Confidence:** 5

**Summary:**

This paper proposed a discrete multi-view clustering method named Discrete Tensorized Label Learning with Anchor Graph, which addresses the interpretability problem by label transmission and maximizing the nuclear norm. Extensive experiments have verified the superiority of the proposed method compared with the state-of-the-art methods.

**Strengths:**

1. The article is well-organized and easy to understand.
2. The idea of maximizing the nuclear norm is somewhat interesting, the theoretical analysis is also clear.

**Weaknesses:**

1.	Innovative may not be enough for such a conference, this work is an expansion of the work[1], and more importantly, why is this work [1] not being compared or even cited? So how do we know the validity of your proposed nuclear norm maximization term?
2.	Many tensor-based methods only work well on ordered datasets, and most datasets have achieved satisfactory performance, like [2],[3]. So what are the advantages of your method, does your algorithm also only work well on ordered datasets? Can your method well handle shuffled datasets?
3.	The datasets are a bit too small and most of the datasets have already achieved the best performance on many tensor-based methods e.g. MSRCv1 and HW, more challenging datasets or datasets with larger sample sizes should be added.
4.	Lack of ablation experiments, from previous tensor methods, tensor methods reach good performance mainly because of the introduction of low-rank tensor modules, so it is difficult to convince the validity of your proposed nuclear norm maximization term because tensor Schatten p-norm has already been shown to have good results. You should design experiments to prove the effectiveness of your proposed nuclear norm maximization term.
5.	Too little experimental analysis content, the authors should consider curtailing the presentation of the comparison algorithm and enriching the experimental analysis section.
6.	Where is complexity analysis? It is very important to have a complexity analysis, which is, after all, the fatal weakness of the tensor approach.
7.	How did you choose the anchors and how did you ensure consistency, in the Scene dataset in Figure 2, the best anchoring rate was achieved at 0.9, which is so different from other datasets, why didn't the authors analyze that phenomenon?
8.	The biggest effect of anchors is to reduce complexity, and the experimental part is not demonstrated. There are too many things missing from the experimental section, and I think the authors should reconsider how to enhance the experimental section. Authors should add ablation experiments, complexity analyses, more datasets, should reduce unnecessary content such as Comparative algorithms because of the small size of the conference paper, should focus on providing important content, and convincing content.

[1] Lu, Han, et al. "Centerless multi-view K-means based on the adjacency matrix." Proceedings of the AAAI Conference on Artificial Intelligence. Vol. 37. No. 7. 2023.
[2] Ji, Jintian, and Songhe Feng. "Anchor structure regularization induced multi-view subspace clustering via enhanced tensor rank minimization." Proceedings of the IEEE/CVF International Conference on Computer Vision. 2023.
[3] Long, Zhen, et al. "S2MVTC: a Simple yet Efficient Scalable Multi-View Tensor Clustering." Proceedings of the IEEE/CVF Conference on Computer Vision and Pattern Recognition. 2024.

**Questions:**

See Weaknesses

---

### Official Review · Reviewer_Bg13 · 2024-11-01

**Soundness:** 2
**Presentation:** 2
**Contribution:** 2
**Rating:** 5
**Confidence:** 5

**Summary:**

This paper improves discrete multi-view clustering by addressing interpretability and balance issues in anchor graph-based methods. It employs label transmission from anchors to samples for enhanced clarity and introduces a novel application of the nuclear norm, proving that maximizing it ensures balanced clusters. The authors propose a new optimization method using first-order Taylor expansion for the nuclear norm and utilize the tensor Schatten p-norm to leverage spatial structure and complementary information between views.

**Strengths:**

1. The proposed method reduces computational complexity by constructing anchor graphs.

2. The method enhances interpretability by using label transmission from anchors to samples, providing clearer clustering results.

3. The optimization process of the algorithm is detailed.

**Weaknesses:**

1. Some fundamental concepts are ambiguously defined, such as when the authors claim that previous methods using graph decomposition or spectral clustering to obtain final clustering labels lack interpretability. What exactly do they mean by "interpretability" in this context? Furthermore, how does this paper specifically address the issue of interpretability?

2. The novelty of this paper is questionable. The authors claim to have discovered a significant application for nuclear norm regularization, stating that maximizing it ensures balanced clustering. However, since the nuclear norm appears with a negative sign in the objective function, this effectively means that the focus is on minimizing the nuclear norm loss term, which is fundamentally similar to methods found in existing literature. Moreover, the use of the Schatten p-norm is also common in previous studies, suggesting that the overall novelty of this paper may be limited.

3.  The paper contains numerous grammatical errors. For instance, lines 362 and 368 employ different tenses, an issue that is also apparent in other parts of the paper.

4. Section 3 emphasizes Definition 3.2, but this is a method proposed in previous literature and should be placed in the related work section instead.

5. The layout of some sections is not well-organized. For example, Section 4.3 is titled "Results and Analysis," but it includes both experimental results and an introduction to comparative methods.

6. The tables in the appendix should be included in the main text.

**Questions:**

See Weaknesses

**Details Of Ethics Concerns:**

N.A.

---

### Official Review · Reviewer_TJKf · 2024-11-02

**Soundness:** 2
**Presentation:** 3
**Contribution:** 2
**Rating:** 3
**Confidence:** 5

**Summary:**

The paper introduces a multi-view clustering method using discrete tensorized label learning with anchor graphs. The approach aims to address common limitations in multi-view clustering by enhancing label interpretability and reducing computational complexity through anchor graph decomposition. Additionally, it proposes the use of nuclear norm maximization for balanced clustering and the tensor Schatten p-norm for spatial and complementary information alignment across views.

**Strengths:**

Reduced Complexity: The anchor graph-based clustering approach significantly reduces computational complexity, making it feasible for large datasets.
Comprehensive Evaluation: Extensive experiments with multiple clustering metrics demonstrate the approach’s effectiveness compared to other state-of-the-art methods.

**Weaknesses:**

Lack of Novelty: Both the Discrete Tensorized Label Learning and Anchor Graphs techniques are well-known, making the novelty of the research insufficient for a top-tier conference.
Limited Innovation in Methodology: The paper mainly combines existing techniques rather than proposing a fundamentally new approach.
Inadequate Real-World Application Examples: The experimental setup lacks a real-world application analysis, which would strengthen the practical impact of the research.

**Questions:**

Could the authors explain how this method would perform on datasets with highly heterogeneous views where spatial structures are vastly different?
Does the model require any specific preprocessing for datasets that are larger and noisier?

**Details Of Ethics Concerns:**

See weaknesses.

---

### Official Review · Reviewer_yoLv · 2024-11-02

**Soundness:** 2
**Presentation:** 2
**Contribution:** 2
**Rating:** 3
**Confidence:** 5

**Summary:**

This paper proposes a tensor-based multi-view clustering method, which attempts to convert the anchors' labels to the samples' labels with certain interpretability and achieve better clustering performance. Experiments on serval datasets are conducted to verify the effectiveness of the method.

**Strengths:**

1. The paper provides several mathematical expressions that effectively convey the main idea behind the proposed method.

2. Experimental results across multiple datasets demonstrate that the proposed method outperforms the other comparative methods.

**Weaknesses:**

1. The motivation and main methodology of the paper appear inconsistent. For instance, while Theorem 3.1 discusses "Unevenly Balanced Cluster Distributions," the paper fails to clarify the relationship between these distributions and the final objective function proposed in Equation (12).

2. The paper does not clearly specify the dimensions of the tensors utilized. Given that FFT is employed to transform the tensor into Fourier space, it is important to note that the results of the FFT are significantly influenced by the arrangement of samples within the tensor. Some previous tensor-based approaches group samples from the same clusters to formulate feature matrices, which subsequently lead to clustering results. However, in real-world applications, samples are often arranged randomly. How do the authors address this challenge when performing FFT?

3. What measures are taken to ensure the optimality of the solution obtained for the discrete variables G?

**Questions:**

As demonstrated in Weaknesses.

---

### Note · Authors · 2025-01-23

I have read and agree with the venue's withdrawal policy on behalf of myself and my co-authors.